# Transparency of Artificial Intelligence in Healthcare: Insights from Professionals in Computing and Healthcare Worldwide

Jose Bernal [1,2,3] and Claudia Mazo [4,*,†]

1. German Center for Neurodegenerative Diseases (DZNE), 39120 Magdeburg, Germany
2. Institute of Cognitive Neurology and Dementia Research, Otto-von-Guericke University Magdeburg, 39120 Magdeburg, Germany
3. Centre for Clinical Brain Sciences, The University of Edinburgh, Edinburgh EH16 4SB, UK
4. DCU School of Computing, Dublin City University, D09 DXA0 Dublin, Ireland
* Correspondence: claudia.mazo@dcu.ie
† Current address: UCD School of Computer Science, University College Dublin, D04 V1W8 Dublin, Ireland.

**Abstract:** Although it is widely assumed that Artificial Intelligence (AI) will revolutionise healthcare in the near future, considerable progress must yet be made in order to gain the trust of healthcare professionals and patients. Improving AI transparency is a promising avenue for addressing such trust issues. However, transparency still lacks maturation and definitions. We seek to answer *what challenges do experts and professionals in computing and healthcare identify concerning transparency of AI in healthcare?* Here, we examine AI transparency in healthcare from five angles: interpretability, privacy, security, equity, and intellectual property. We respond to this question based on recent literature discussing the transparency of AI in healthcare and on an international online survey we sent to professionals working in computing and healthcare and potentially within AI. We collected responses from 40 professionals around the world. Overall, the survey results and current state of the art suggest key problems are a generalised lack of information available to the general public, a lack of understanding of transparency aspects covered in this work, and a lack of involvement of all stakeholders in the development of AI systems. We propose a set of recommendations, the implementation of which can enhance the transparency of AI in healthcare.

**Keywords:** transparency; healthcare; artificial intelligence; international survey; interpretability; privacy; security; equity; intellectual property

## 1. Introduction

Artificial Intelligence (AI)-based technologies are becoming increasingly common in our daily lives. In healthcare, AI holds great promise for supporting clinical decision-making by facilitating complex, impractical, or time-consuming duties [1–4], including prediction [5–11], diagnosis [12–17], treatment [18–20], and follow-up [21–23]. All of these developments will undoubtedly revolutionise healthcare in the coming years [24].

The immense potential of AI in healthcare has also begun to raise serious ethical and legal concerns [4,25–27]. Healthcare professionals may hesitate, for example, to use the most powerful AI models as these are "black boxes", the decisions and recommendations of which cannot be fully understood or explained most of the time—not even by computing specialists. Furthermore, establishing the responsibility of AI systems whenever they make mistakes—e.g., due to algorithmic bias—may be difficult, as are the accompanying legal actions [4]. It is difficult for these systems to acquire the trust of healthcare professionals and patients and translate them into clinical practice without significant compromise and preparedness of all stakeholders [4,25–31].

Transparency has thus become an urgent and pressing challenge that must be addressed to combat the public's lack of trust in AI [4,32]. This complex and evolving term

"transparency" encompasses, at its core, issues surrounding the use of AI, such as interpretability, privacy, security, equity, and intellectual property [33]. In this work, we understand transparency from these five viewpoints.

The scientific community's interest in AI transparency in healthcare is growing at an exponential rate (Figure 1). Aside from recognising the potential and promise that AI has for healthcare, literature reviews in specific study fields are increasingly focusing on outlining barriers that must be surmounted before AI can be reliably deployed and making recommendations to make this happen. Robin et al. [17] argue that communication problems among stakeholders (rationalised in that work in terms of computing and healthcare experts) render the majority of work inaccessible to end users and unlikely to develop further and discuss how misalignment between open research and the industry's goal in intellectual property protection contributes to the lack of reproducibility and openness. To cope with this situation, they call for engaging clinicians, patients, and relatives in the research process; working towards better representation of global ethnic, socioeconomic, and demographic groups for mitigating equity issues that may end up harming already marginalised and underrepresented groups; and creating and promoting open science initiatives that lead to better reporting, data sharing, and reproducibility, e.g., Human Connectome Project [34]. Mazo et al. [31] provide evidence of the lack of cancer research translation into clinical practice, emphasising that this step remains difficult, in part due to concerns regarding validation and accountability, given that medical mistakes produced by biased or defective AI may threaten patients' lives. On a more optimistic note, the authors note that two projects, the Papanicolau test and Paige Prostate for cervical cancer screening and prostate cancer risk prediction, have recently acquired Food and Drug Administration (FDA) regulatory clearance, a huge step towards real-life deployment. Sajjadian et al. [19] found that, despite the promise of AI in predicting (depression) treatment outcomes, the methodological quality of AI-based investigations is heterogeneous and that those presenting overly optimistic findings are often characterised by inadequate validation. In light of this observation, the authors recommend a more complete assessment of generalisability and replicability, and they advocate using tutorials, consensus criteria, and a checklist to design more effective and deployable AI systems for such purposes.

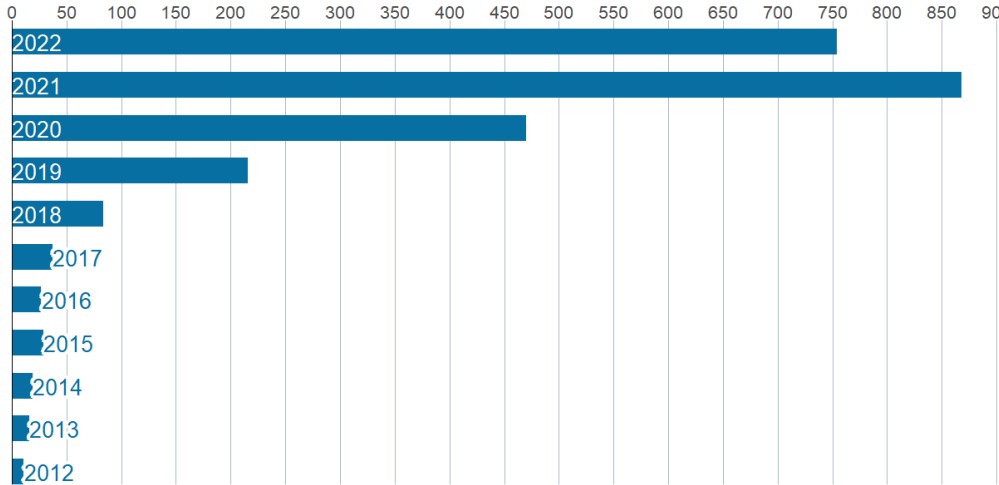

**Figure 1.** Number of peer-reviewed journal articles discussing AI transparency in healthcare. Using the Web of Science, we searched and counted the works containing (Transparency, Ethics, Equity, Interpretability, Explainability, Security, Privacy, Intellectual property, Trust, Reliability) and (Healthcare, Medicine, Diagnosis, Prognosis, Prediction, Health, Care, Medical) and (Artificial intelligence) in their abstracts.

The common denominator in these and other studies is that there are serious transparency problems that have yet to be addressed, as well as a mismatch between academics' desire to publish research papers showcasing sophisticated AI techniques versus the actual application of such research in healthcare. Stakeholder perspectives on AI trust have also come to the forefront of the literature [28,35,36]. Martinho et al. [28] surveyed 77 medical doctors in The Netherlands, Portugal, and the U.S. about the ethics surrounding Health AI. They found that physicians (i) recognise the need to automate elements of the decision-making process but also state that they must be the ones making the final decision, (ii) acknowledge that, while private tech companies seek profit, it is fundamental to assess AI to the highest standards and hold corporations responsible for any harm caused by their tools, and (iii) highlight the importance of engaging doctors—as end-users— throughout the development process. Esmaeilzadeh [35] collected and analysed responses from 307 individuals in the USA about their perceptions of AI medical devices with clinical decision-support characteristics. The authors found that higher concerns, especially regarding performance anxiety and communication barriers, coincide with higher perceived risk and lower benefit, implying that when patients do not comprehend how AI devices work, they distrust them the most. The authors advise carrying out and publishing studies on the benefits and drawbacks of using AI devices such that customers are well aware of them and may have more trust in AI.

Identifying the challenges that professionals in computing and healthcare perceive with AI in healthcare practice might serve as a springboard for additional education, research, and policy development [4,30]. Thus, the question this paper seeks to answer is: *what challenges do experts and professionals in computing and healthcare identify concerning transparency of AI in healthcare?* Here we respond to this question based on recent literature discussing this topic, and on a global online survey, we sent to professionals working in computing and healthcare and potentially within AI. Based on these, we propose a set of recommendations to enhance AI transparency in healthcare.

## 1.1. Background

Transparency in AI refers to algorithms that are expressive enough to be human intelligible on their own or when used in conjunction with external tools. This entails providing stakeholders with pertinent information about the entire process, including, but not limited to: whole process documentation, disclosure of training and validation procedure properties, dataset description, data analysis, code releases, and model and results explanation [37]. In this paper, we look at AI transparency in healthcare from five corners: interpretability, privacy, security—General Data Protection Regulation (GDPR), equity, and intellectual property. We define these concepts in the following sections.

### 1.1.1. Interpretability

The capacity of an AI system to describe its decision-making process is known as interpretability. This ability is connected to how effectively people can grasp and follow a certain choice and procedure and recognise when the model has made a mistake [38,39]. It is in everyone's best interest to have an interpretable system. Data scientists would be able to communicate their models' findings to their target audience. Healthcare practitioners would understand why and how such models arrive at a certain decision or recommendation; they would be sure that these AI systems adhere to medical guidelines, and be aware of the risks these may bring to patients [40,41]. End-users would have the confidence that these systems will provide decisions that meet the maximum standards. Business stakeholders, regulators, and lawmakers would be able to safeguard end-users by enforcing transparency.

### 1.1.2. Privacy

Data privacy is concerned with the correct treatment of data, such as whether or not personal information is shared with third parties. With the rise of big data, data privacy has sparked a heated discussion since it is both necessary and dangerous. On the one hand, data availability provides the foundation for improved AI system performance. Without enough training data, AI models are potentially biased and unable to generalise to unseen datasets. On the other hand, careless release and handling of private data is extremely risky and can result in privacy breaches, trustworthiness, penalties, or civil suits [42]. The United States Health Insurance Portability and Accountability Act (HIPAA) [43], the Electronic Communications Privacy Act (ECPA) [44], the Children's Online Privacy Protection Act (COPPA) [45], and the EU's GDPR [46] are just some examples of data privacy regulations throughout the globe.

### 1.1.3. Security

Data protection security refers to any procedures that use suitable technological or organizational methods to safeguard digital information from unauthorized or illegal access, corruption, destruction or damage, or theft across its full life cycle [46]. The GDPR is the world's strongest privacy and security regulation. Despite the fact that it was designed and passed by the EU, it puts duties on all enterprises that target or collect data about people in the European Union [46].

### 1.1.4. Equity

Population representativeness within the training dataset determines the generalisation power of an AI system [47–49]. If an AI method employs data gathered in an inequitable manner, the model and its decisions will be biased and have the potential to harm misrepresented groups. Thus, data equity focuses on acquiring, processing, analysing, and disseminating data from an equitable viewpoint, as well as recognising that biased data and models can perpetuate preconceptions, worsen racial bias, or hinder social justice [49,50].

### 1.1.5. Intellectual Property

Intellectual property refers to the ownership of inventions that can have moral and commercial worth and legal protection [51]. Such protection aims to ensure benefits generated from the exploitation of an idea or product benefit society and the inventor. Individuals and institutions with intellectual property have the right to bar others from using their inventions, having a direct and significant influence on their use or sale [51].

## 2. Material and Methods

### 2.1. Online International Survey

We conducted a short survey of professionals working in computing and healthcare and potentially within AI to support the claims and recommendations made in this article.

### 2.1.1. Data Collection

Using the "Google Forms" platform, we created an anonymous electronic survey on AI transparency in healthcare. We discussed preliminary drafts with six professionals from the computer science and healthcare fields. We included all of their suggestions in the final version. We circulated this survey to professionals working in computing and healthcare and potentially within AI through a list of social networking and external collaborators by social networks. We conducted it between 16 July 2021 and 1 November 2021. We aimed for a sample size of 40 respondents due to the exploratory nature of this survey and the survey fatigue effect experience during the COVID-19 pandemic [52,53].

Each participant was given a random and unique link to the online survey. The questionnaire's preamble and introduction both offered information on the survey's objective. We excluded healthcare experts who had no prior experience with AI systems, as well as computer science professionals who used AI but not for medical purposes.

The survey was approved by University College Dublin's Research Ethics Committee (ref. LS-21-42-Vargas; and in accordance with the Declaration of Helsinki. The survey does not include any sensitive or identifiable data from the participants in agreement with the GDPR [46]. The Google Form did not record any responses unless participants pressed the "submit" button at the end of the questionnaire. Moreover, we configured the form to allow a single submission per participant. Informed consent was implied once the "submit" button was pressed.

### 2.1.2. Aspects of Interest

The online survey questions are condensed in Table 1. We devised it in accordance with the checklist for reporting the results of internet e-surveys [54]. We also included definitions for each survey section using generic and clear language to avoid ambiguities with terms potentially unfamiliar to some respondents. Questions spanned six categories: participant demographics, interpretability, privacy, security, equity, and intellectual property. All of these questions were required. Participants who responded "no" in Q12, Q18, Q22, Q23, and Q26, or "yes" in Q26 were asked a follow-up question about why they answered that.

### 2.1.3. Data and Statistical Methods

We stored data securely until the end of the study and archived it afterwards. We used descriptive statistics to analyse survey responses and collected and summarised open-text comments—i.e., Q12, Q18, Q22, Q23, and Q26.

**Table 1.** Online questionnaire.

| Section | Question | Answer |
|---|---|---|
| Survey participants | Q1. What country do you live in? | Open text |
| | Q2. How old are you? | [18–25]; (25–35]; (35–45]; (45–55]; >55 |
| | Q3. To which gender identity do you most identify? | Man; woman; non-binary; prefer not to disclose; prefer to self-describe |
| | Q4. What is your profession? | Physician; industry worker; researcher; teacher; graduate student; undergraduate student; other (open text) |
| | Q5. What is your area of expertise? | Computer science; morphological science; physiopathology; neuroanatomy; embryology; histology; anatomy; neurology; radiology; other (open text) |
| | Q6. How many years of experience do you have in your area of expertise? | [0–5]; (5–10]; (10–15]; >15 |
| | Q7. Do you use AI systems routinely? | Yes; no |
| | Q8. How many AI systems have you used? | [0–2]; (2–5]; (5–7]; (7-10]; >10 |
| Interpretability | Q9. Do AI systems that you use routinely explain to you how decisions and recommendations were made (underlying method)? | Yes, I clearly understand the underlying method embedded in the AI system; yes, I superficially understand the underlying method embedded in the AI system; no, I do not know anything about the underlying method embedded in the AI system |
| | Q10. Do AI systems that you use routinely explain to you why decisions and recommendations were made (medical guideline)? | Yes, I am sure that the medical guideline used within the AI system is appropriate; no, I am not sure that the medical guideline used within the AI system is appropriate |
| | Q11. Can the AI system be interpreted so well the end user can manipulate it to obtain a desired outcome? | Yes, the AI system is explainable enough to the level where end users can influence it to produce desired; no, even though the AI system is explainable enough, the system is robust enough to avoid this problem; no, end users only have access to the final decision or recommendation |
| | Q12. In general, are your needs regarding interpretability addressed adequately by AI systems? | Yes, I agree; no; if not, why not? |
| Privacy | Q13. Do you know the patient population characteristics used for training and validating the AI system? | Yes, I know or have access to the patient population characteristics used for training and validating the AI system; no, I do not know or have access to information concerning AI system training or validation |
| | Q14. Do you know what equipment, protocols, and data processing workflows were used to build the training and validation dataset? | Yes, I know or have access to the afore information; no, I do not know or have access to the afore information |
| | Q15. Do you know who annotated or labelled the datasets used for training and validating the AI system (credentials, expertise, and company)? | Yes, I am fully aware of the team who phenotyped or annotated the training and validation datasets; no, I am not aware of who carried out such a process |
| | Q16. Are both training and validation datasets available to the general public or via request? | Yes, both datasets are available to the general public or via request; no, data are available, but annotations are not; no, one of these datasets is not available; no, neither of these datasets is available; I am not aware of this information |

Table 1. *Cont.*

| Section | Question | Answer |
|---|---|---|
| | Q17. In general, are datasets typically available in your area of expertise? | Yes, datasets are often available to the general public in my area of expertise; yes, datasets are private but they may be accessed after approval; no, datasets are typically private and difficult to access |
| | Q18. In general, are your needs regarding privacy addressed adequately by AI systems? | Yes, I agree; no; if not, why not? |
| Security - GDPR | Q19. Does the AI system you have been in contact with require any data storage service? | No; yes, an internal data storage service; yes, an external data storage service |
| | Q20. Does the data storage service used by the AI system contemplate secure data access (e.g., user hierarchy and permissions)? | No, all users have full permission to write, read, modify files within the data storage; yes, an internal information technology team manages secure data access; yes, an external manages secure data access |
| | Q21. Who is responsible if something goes wrong concerning AI system security? | The person causing the security breach; the principal investigator or line manager; the information technology team; the AI system manufacture company; the institution which bought the AI system; I am not aware of who would be responsible in such a case; other (open text) |
| | Q22. In general, are your needs regarding security addressed adequately by AI systems? | Yes, I agree; no; If not, why not? (open text) |
| | Q23. In general, are the needs of the patients regarding security addressed adequately by AI systems? | Yes, I agree; no; if not, why not? (open text) |
| Equity | Q24. Are the training and validation datasets biased in such a way it may produce unfair outcomes for marginalised or vulnerable populations? | Yes, I am aware of data bias which may potentially discriminate marginalised or vulnerable populations; no, neither presents data bias that may potentially discriminate marginalised or vulnerable populations; no, I am not aware of such information |
| | Q25. Are the training and validation datasets varied enough (e.g., age, biological sex, pathology) so that they are representative of the people for whom it will be making decisions? | Yes, both datasets are varied enough to be considered representative of the target population; no, both datasets are not varied enough to be considered representative of the target population (e.g., low sample sizes or imbalance datasets); no, I am not aware of such information |
| Intellectual property | Q26. In general, do you think there is the right balance between intellectual property rights and AI transparency/fairness goals? | Yes; no; I do not have enough information to judge this aspect; if yes/no, why/why not? |
| | Q27. In general, should the data used to train and validate an AI system be disclosed and described in full? | Yes; no; I do not have enough information to judge this aspect; if yes/no, why/why not? |

## 3. Survey Results

### 3.1. Sample Characteristics

A total of 40 professionals from nine countries in North and South America, Europe, and Asia completed the survey (50% women). Collected demographics are condensed in Figure 2. The majority of participants were between the ages of 26 and 45 (70%), had between zero and ten years of experience in their fields (65%), and use AI systems routinely (67.5%). Areas of expertise were either computer science (67.5%) or healthcare (32.5%)—including morphological science, physiopathology, neuroanatomy, embryology, histology, anatomy, neurology, radiology, neuropsychiatry, pathology, biologist, and biomechanics. Most participants used between zero and five AI systems habitually (62.5%).

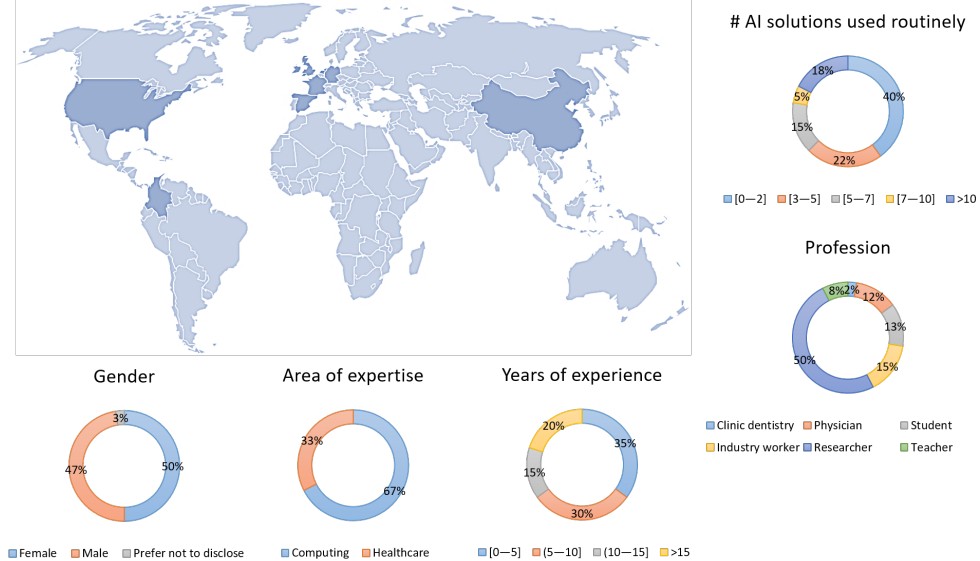

**Figure 2.** Survey sample characteristics. A total of 40 professionals from nine countries in North and South America, Europe, and Asia completed the survey.

### 3.2. Interpretability

Q9  *Do the AI systems that you use routinely explain to you how decisions and recommendations are made (underlying method)?*  Only 27.5% of participants, all of whom have a computer science background, answered AI systems explained how decisions and recommendations were made.

Q10  *Does the AI systems that you use routinely explain to you why decisions and recommendations are made (medical guideline)?* The majority of participants (57%) are unsure of whether AI systems adhere to medical guidelines.

Q11  *Can the AI system be interpreted so well that end users could manipulate it to obtain certain outcome?* Most respondents (60%) stated that because end users only have access to the final choice, it is difficult for them to trick AI systems on their own. However, around 17.5% of participants feel AI systems can be understood to the point where end-user manipulation is possible.

Q12  *In general, are your needs regarding interpretability addressed adequately by AI systems?* The great majority of respondents agreed interpretability demands have been met thus far (yes: 75%; no: 25%), arguing that "giving more information to end users will be confusing and overall, time-consuming". Those in disagreement recognised interpretability is still a "work in progress" but that "simpler explanations", including "literature, references, or step by step", should be displayed to professionals along with the final response.

### 3.3. Privacy

Q13 *Do you know the patient population characteristics used for training and validating?* A considerable proportion of respondents (40%) is unaware of patient population characteristics used for training and validating AI systems they use on a regular basis or are unable to access such information.

Q14 *Do you know what equipment, protocols, and data processing workflows were used to build the training and validation dataset?* Answers to this question were even: around half of the participants manifest to know or have access to such information, and the remaining half that they do not.

Q15 *Do you know who annotated or labelled the datasets used for training and validating the AI system (credentials, expertise, and company)?* Around 58% of participants are not aware of the team in charge of this process.

Q16 *Are both training and validation datasets available to the general public or via request?* Only a fourth of the participants are aware of AI systems whose training and validation datasets are available to the general public; the remainder were evenly split between not knowing whether such data were available (37.5%) and knowing such data were either partially or completely unavailable (37.5%).

Q17 *In general, are datasets typically available in your area of expertise?* More than half of participants stated datasets tend to be unavailable to the general public (80%), although a few private ones can be accessed after approval (20%).

Q18 *In general, are your needs regarding privacy addressed adequately by AI systems?* According to survey respondents, their privacy demands are properly met by the AI system they typically employ (yes: 90%; no: 10%). Nonetheless, some expressed their concerns about data availability, the lack of specifics about how AI systems were built and evaluated, and the loss of control when employing cloud-based technology owing to privacy regulations.

### 3.4. Security

Q19 *Does the AI system you have been in contact with require any data storage service?* For AI systems that survey participants use on a regular basis, no storage or internal data storage is usual (none: 25%; internal: 70%; external: 5%).

Q20 *Does the data storage service used by the AI system contemplate secure data access (e.g., user hierarchy and permissions)?* Although in most cases, participants manifested an internal information technology team managed data access ($\approx$ 58%), about a third of them agreed with the following statement: "all users have full permission to write, read, and modify files within the data storage".

Q21 *Who is responsible if something goes wrong concerning AI system security?* The responsibility on whom security issues fall is not sufficiently clear to many survey participants (40%). Only 5% of participants would blame the AI system for data breaches, and the rest would make responsible someone within their institution: the person causing the breach (10%), line manager or principal investigator (25%), information technology team (5%), the institution which acquired the AI system (12.5%), or all of the above (2.5%).

Q22 *In general, are your needs regarding security addressed adequately by AI systems?* The vast majority of respondents indicate security needs are currently met (95%). Even though some answered positively to this question, they rely on "team awareness" and recognise that "security breaches are always possible".

Q23 *In general, are the needs of the patients regarding security addressed adequately by AI systems?* The great majority of survey participants (93%) answered their security demands are addressed thus far. Those disagreeing manifest "security breaches are always possible" and that additional "protection besides patient anonymisation" would be required to minimise security problems.

### 3.5. Equity

Q24 *Are the training and validation datasets biased in such a way that they may produce unfair outcomes for marginalised or vulnerable populations?* About half of survey participants believed that bias during training and validation of the AI systems could produce unjust outcomes for marginalised or vulnerable groups (yes: 47.5%; no: 15%; unaware: 37.5%).

Q25 *Are the training and validation datasets varied enough (e.g., age, biological sex, pathology) so that they are representative of the people for whom it will be making decisions?* A fourth of participants indicated that representativeness in training and validation datasets is lacking and a third are unaware of such details (yes: 40%; no: 25%; unaware: 35%).

### 3.6. Intellectual Property

Q26 *In general, do you think there is the right balance between IP rights and AI transparency/fairness goals?* Most respondents do not have enough information to determine whether there is a proper balance between IP rights and transparency (58%). Participants expressed the following concerns: "The big problem with IP is that individuals are cosmetically involved; generally, the good part of the IP goes to the institutions. This remains true for all research-derived products, including AI" and "I think the community is very open to sharing advances and new discoveries and results. These are two things on the two sides of scales. More IP rights leads to less transparency, and vice-versa".

Q27 *In general, should the data used to train and validate an AI system be disclosed and described in full?* The great majority of survey participants agreed information concerning training and validation needs to be described in full (yes: 70%; no: 12.5%; not confident to judge this aspect: 17.5%). Participants manifested this situation would enable "reproducibility" and "repeatability" in research, and professionals using an AI system can detect "any possible bias" or "limitation" and "be more confident about [using] it". However, some pointed out the importance of "sharing as much as possible" without violating the "GDPR regulations".

### 3.7. General Perceptions by Area of Expertise and by Years of Experience

To further understand whether there were different perceptions depending on the participants' background or experience, we stratified their responses to Q12, Q17, Q18, Q22, Q23, and Q27 by these two factors (Figure 3a by area and Figure 3b by experience).

We anticipated that when examining replies by area of expertise, we would see different patterns depending on whether the participants came from a clinical or technological background. However, this was not really the case as both of these groups exhibited similar trends (Figure 3a): they see privacy and security favourably (Q18, Q22, Q23), have mixed views on interpretability and reporting (Q12 and Q27), and perceive dataset availability and balance between IP rights and transparency unfavourably (Q17 and Q26).

We expected to see experienced professionals with a less favourable opinion of AI in healthcare versus those with less experience due to the relative novelty of AI and its technological, methodological, and ethical requirements and implications for clinical practice. Nonetheless, this does not seem to be the case either (Figure 3b): participants with more expertise indicated that existing AI solutions met their demands for privacy, security, and patient security (Q18, Q22, and Q23) in general, but those with fewer years of experience were slightly more doubtful. Furthermore, individuals with fewer years of expertise criticised AI more regarding interpretability, dataset availability, the balance of IP rights and openness, and reporting (Q12, Q17, Q28, and Q27) than the more experienced ones. A relationship between perception and years of experience does not appear to exist for these four elements.

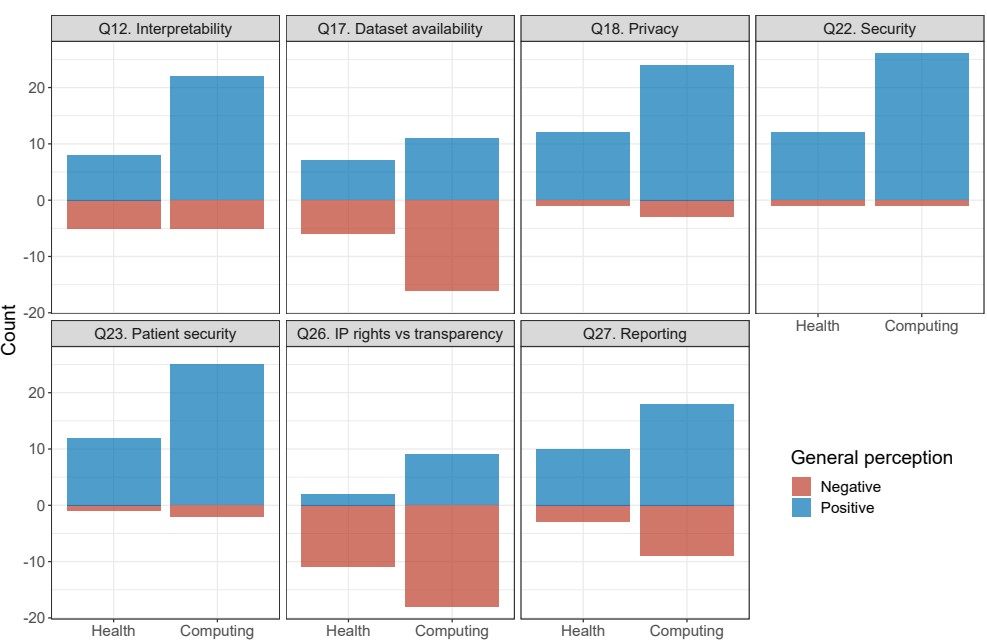

(**a**) General perception by area of expertise

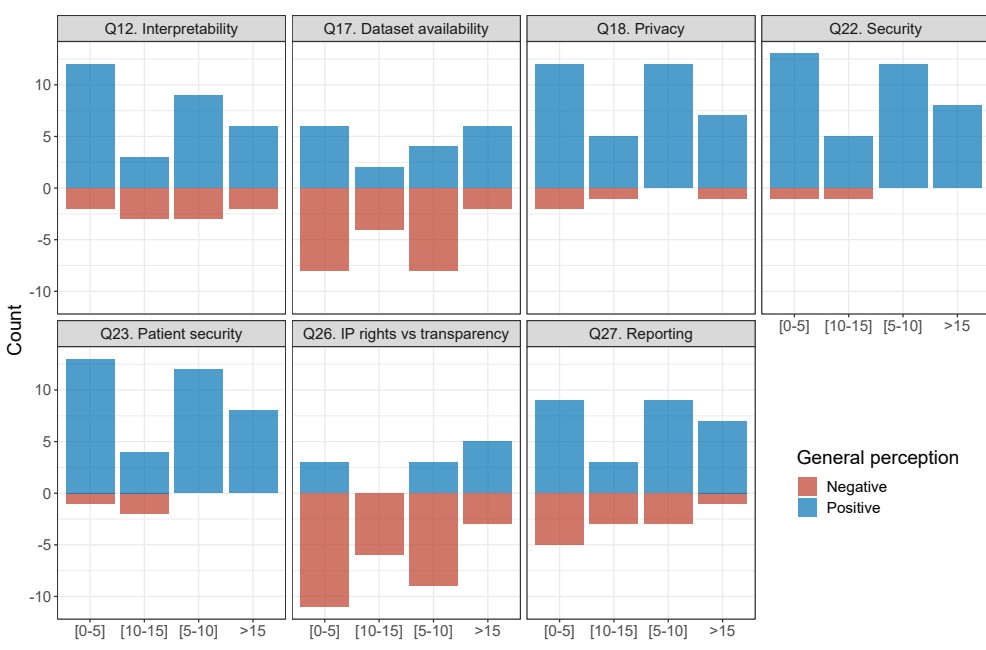

(**b**) General perception by years of experience

**Figure 3.** General perception of the survey respondents stratified by area of expertise and by years of experience.

Regardless of expertise and years of experience, there are unmet expectations regarding interpretability, dataset availability, balance between IP rights and transparency, and reporting (Q12, Q17, Q26, and Q27). Worryingly, our findings are consistent with and backed up by ongoing conversations [17,31,35,36,55].

## 4. Discussion

### 4.1. Interpretability

Because AI-based healthcare systems may have an impact—either positive or negative—on people's lives, it is fundamental to dispel the myths around these black boxes in order to

increase trustworthiness [56,57]. There are some works that focus on specific AI techniques in healthcare and interpretability, such as common methods, features, training and testing methodologies, metrics, and current challenges relating to implementation in clinical practice (these aspects could vary in terms of the project aim, disease, and input/output sources) [30,31]. It is, consequently, no surprise that interpretability has become a prominent subject in AI for healthcare applications in recent years [39,58–62]. However, it remains an open problem due to current theoretical gaps between model complexity and explainability; insufficient feedback from all stakeholders throughout the development process (e.g., via evaluations of user satisfaction, explanation goodness, acceptance, and trust in AI-based systems) [17,35]; and inadequate traceability indicating how models reach conclusions [39, 63]. According to our international survey, a fundamental issue that needs to be addressed is the unequal understanding among professionals: most healthcare professionals do not understand how AI systems operate nor whether these adhere to medical guidelines. However, participants also manifested that the level of granularity is essential. While a certain degree of clarity of the whole process—equations, biography, medical guideline, and algorithms—seems reasonable, providing excessive details may be confusing and of no practical value. We propose the following recommendations for better interpretability:

- *Involvement of all stakeholders throughout all phases of the software development life cycle*. The development of a healthcare system leveraging AI is a multidisciplinary process that requires continuous communication between system creators and those affected by their use (e.g., healthcare professionals, policy-makers, and patients) [17,64]. This ensures explanations are clear and sufficient, real interpretability needs are addressed timely, and limitations are known by the target population.
- *Interpretability over complexity*. Complexity is entangled with interpretability: predictions made by a complex model are more difficult to comprehend and explain than a simpler one. Since trust in AI systems can only be improved by improving interpretability, the use of simpler yet interpretable methods should be considered for healthcare applications over black boxes [39,64].
- *Uncertainty quantification*. Real-world data are far from perfect; they contain missing values, outliers, and invalid data. Inputting it into models trained and tested on well-selected datasets may result in flawed verdicts [65]. Quantifying uncertainty for a certain input–output pair thus allows developers and users to understand whether they can trust a model's prediction, whether the cause of the problem is in the input data or the AI system's inability to handle such a case. It should be noted that the latter could be employed to improve model robustness.
- *Evaluation beyond accuracy*. "[. . . ] AI research should not be limited to reporting accuracy and sensitivity compared with those of the radiologist, pathologist, or clinician" [66]. Performance evaluation should be accompanied by a thorough and multidisciplinary assessment of interpretability [67]. Evaluations striving to explain the reasoning behind a certain prediction can improve our understanding of healthcare. Assessing interpretability and explainability requires maturation [39].

### 4.2. Privacy

Big data are essential for developing universal and robust AI systems that can be later applied to healthcare purposes [68,69]. Efforts have been undertaken to expand the availability and accessibility of healthcare data. Simultaneously, multiple entities overseeing privacy have been created in an attempt to strike a balance between patient privacy and the ability to exchange patient data for healthcare and research purposes [43–46,69]. According to our international survey, data unavailability, insufficient details on training and validation, and potential risks in external cloud processing or storage platforms are of concern. We propose the following recommendations for improving privacy:

- *Data privacy impact assessments*. The implementation of rigorous data privacy impact assessments within healthcare centres permits finding practices that violate privacy and determining whether additional procedures need to be implemented to protect

patient data further [70]. Likewise, procurement law must ensure all AI systems provided by third parties comply with strict privacy policies [70].

- *Audit trails*. Records of who is doing what, what data are being utilised/transmitted, and what system modifications are being performed must be retained, for example, by means of audit trails [70].
- *Cloud platforms and privacy*. Cloud platforms are excellent solutions for reducing operational costs and increasing productivity (processing speed, efficiency, and performance). At the same time, they require transmitting data from healthcare centres to an external location, thereby increasing the likelihood of tracking sensitive and private information [42,71]. Appropriate strategies to regulate data access, anonymisation prior to transmission and storage are critical to ensure the privacy of patients is not put at risk.
- *Information availability*. Information regarding training and testing population characteristics, data acquisition protocols and equipment, and team expertise must be made clear and available to stakeholders since these permit determining potential limitations of the AI system. For example, evidence AI systems that have been tested on multiple and heterogeneous datasets need to be handed over by providers as proof it generalises well and can be safely deployed into healthcare practice.

### 4.3. Security

In the healthcare area, digital records are replacing paper-based ones, easing medical practice and allowing continual access to patient health information [72,73]. Nonetheless, because patient information is extremely valuable, digitalisation has also made this industry a major victim of external and internal assaults around the world [72]. Clearly, this issue became expanded with the COVID-19 pandemic as more employees transitioned away from office work and toward entirely remote or hybrid work, thereby making the healthcare system even more exposed to external threats. Our survey suggests users are greatly unaware of the aforementioned risks as they perceive the AI systems they use in their routine practice to be safe and comply with GDPR. Moreover, they seem unsure about whose responsibility it is when a data leakage occurs. We propose the following recommendations for improving security:

- *Device encryption*. All devices that employees use to access corporate data should be completely encrypted, and sensitive data stored and transferred in an encrypted way [74].
- *Security testing*. Effort, time, and money should be invested in cyber-security. Hiring a professional organisation to conduct security audits is an excellent way to test data security, as these reveal security weaknesses [75], especially in remote work setups. These professionals, as independent organisations, can verify that AI systems comply with the utmost security standards.
- *Restrict and control access/modification/delete/storage data and device use access*. Our survey showed a third of respondents did not have user hierarchies and permission settings set up in their organisations, increasing the risk of data breaches or mishandling. Employees should only have access to data that is absolutely necessary for them to accomplish their jobs [76].
- *Employee training*. Training all members on why security is important and how they can contribute can not only decrease risks but also improve reaction times when breaches occur [77]. Frequent training sessions and up-to-date policy information can encourage employees to put these guidelines into practice. For example, an easy and common employee mistake is to write down their passwords on a sticky note.

### 4.4. Equity

Ensuring equity guarantees fair and safe outcomes regardless of a patient's race, colour, sex, language, religion, political opinion, national origin, or political affiliation. A plethora of AI-based methods use large amounts of (historical) data to learn outputs for given inputs.

If such data are unrepresentative, inadequate, or present faulty information—e.g., reflecting past disparities—then AI models end up biased. Even if developers have no intention of discriminating against vulnerable or marginalised populations, this bias, when left unchecked, can result in judgments that have a cumulative, disparate impact [78,79]. According to our survey, fortunately, a sizeable portion of healthcare and computing professionals that took part in it are aware of this situation. Nonetheless, a lack of information prevails: about a third of survey participants did not have sufficient data to judge aspects of equity. We propose the following recommendations for improving equity:

- *Release information to the public.* AI system developers must release demographic information on training and testing population characteristics, data acquisition, and processing protocols. This information can be useful to judge whether developers paid attention to equity.

- *Consistency in heterogeneous datasets.* Research conducted around the world by multiple institutions has demonstrated the effectiveness of AI in relatively small cohorts of centralised data [3]. Nonetheless, two key problems regarding validation and equity remain. First, AI systems are primarily trained on small and curated datasets, and hence, it is possible that they are unable to generalise, i.e., process real-life medical data in the wild [2]. Second, gathering enormous amounts of sufficiently heterogeneous and correctly annotated data from a single institution is challenging and costly since annotation is time-consuming and laborious, and heterogeneity is evidently limited by population, pathologies, raters, scanners, and imaging protocols. Moreover, sharing data in a centralised configuration requires addressing legal, privacy, and technical considerations to comply with good clinical practice standards and general data protection regulations and prevent patient information from leaking. The use of federated learning, for example, can help overcome data-sharing limitations and enable training and validating AI systems on heterogeneous datasets of unprecedented size [80].

### 4.5. Intellectual Property

Many people struggle to strike a balance between openness and IP, despite the fact that transparency is a necessary component for effective IP. IP mechanisms bring a period of exclusive rights to the authors in exchange for details about the invention or product, allowing others to build upon such innovation. Nevertheless, patent examiners continue to face challenges in carrying out their duties since most of the data that may assist them is not freely available, owing to the fact that the organizations maintaining such material have not recognized that making it public would be useful [81]. Transparency can help steer the development of a better IP system. We just saw a solid example of this necessity in the form of IP protections on COVID-19 vaccinations. COVID-19 vaccinations were created in an unprecedented amount of time. However, the virus continues to mutate, and new strains emerge, hastening contagions and increasing the death toll. As a result, it is critical to remove the legal impediments to increasing global vaccine manufacturing. The World Health Organization established the COVID-19 Technology Action Pool [82] to foster vaccine technology and know-how sharing, but none of the companies producing vaccines have signed up to date. We propose the following recommendations for improving intellectual property:

- *Invest in the correct guidance early on.* Identifying sensitive data and evaluating the critical importance of protecting it from the public domain should be the first priority. Our survey participants agree greatly with the need for disclosing and describing information related to training and validation in full. This should evidently be accompanied after evaluating what information must be safeguarded and what information may be made available to the public—e.g., by performing a risk and cost-benefit analysis without jeopardizing openness.

- *Educate employees about intellectual property*. Employees should be made aware of the value, boundaries, and risk of IP (or lack thereof) [83–85]. Training should make clear to them what needs to be protected, how to protect it, and from whom it should be protected [83–85].

## 5. Conclusions

Artificial intelligence has the potential to transform healthcare in the near future. However, in order to be converted into actual practice, these systems must acquire the trust of healthcare professionals and patients, which cannot be achieved without enhancing their transparency. Understanding the challenges and potential solutions for this purpose is crucial for informing future policies and practices for educating computing and health professionals. Based on the current literature and an international survey, we conclude that there is an evident need for: engaging healthcare professionals as these systems are implemented in clinical settings and creating new regulations overseeing the transparent design, validation, deployment, and certification of systems leveraging artificial intelligence. Concerns do not seem particularly linked to the background of the respondents or their years of experience.

Findings from our international survey are somewhat limited by the number of respondents ($n = 40$). We attribute this situation to the survey fatigue effect experienced during the COVID-19 pandemic [52,53]. There are two key aspects to highlight nonetheless. First, we received responses from professionals in healthcare and computing leveraging AI in their routine practice from various countries around the globe. Second, even in this relatively small sample, we were able to note key problems regarding the generalised lack of information available to the general public, of understanding of transparency aspects covered in this work, and of involvement of all stakeholders in the development of AI systems. Future work should consider elaborating on these aspects further, either by means of more respondents or with questionnaires focused on fewer yet deeper items of transparency.

Our work is well aligned with emerging discussions of American and European Technology Policy Committees around the trustworthy use of AI in healthcare, e.g., the e-health proposition made by the European Commission in "shaping Europe's digital future" [86]; the remarks on AI-augmented software in embedded medical devices made by the Food and Drug Administration (FDA) [87]; the policy statement on algorithmic transparency and accountability made by the Association for Computing Machinery (ACM) [88]; the urgent health care needs and concerns heightened by the COVID-19 pandemic [89]; and the prospective and retrospective transparency's elements included in the GDPR under Article 5(1)(a).

**Author Contributions:** J.B.: Conceptualization, Methodology, Software, Validation, Formal analysis, Investigation, Writing—Original Draft, Visualization. C.M.: Conceptualization, Methodology, Software, Validation, Formal analysis, Investigation, Data Curation, Writing—Original Draft, Visualization, Project administration. All authors have read and agreed to the published version of the manuscript.

**Funding:** This research received no external funding.

**Institutional Review Board Statement:** The survey was approved by University College Dublin's Research Ethics Committee (ref. LS-21-42-Vargas) and in accordance to the Declaration of Helsinki. The survey does not include any sensitive or identifiable data from the participants, in agreement with the GDPR [46]. The Google Form did not record any responses unless participants pressed on the "submit" button at the end of the questionnaire. Moreover, we configured the form to allow a single submission per participant. Informed consent was implied once the "submit" button was pressed.

**Informed Consent Statement:** Informed consent was implied once the "submit" button was pressed.

**Data Availability Statement:** The datasets generated during and/or analysed during the current study are available from the corresponding author on reasonable request.

**Acknowledgments:** We would like to thank all survey participants for their responses despite the COVID-19 survey fatigue. We would also like to express our gratitude towards Catherine Mooney for her insightful and constructive feedback throughout the development of this work. Claudia Mazo was affiliated with the UCD School of Computer Science at University College, Dublin, at the time of the survey development and is currently affiliated with DCU School of Computing at Dublin City University.

**Conflicts of Interest:** The authors declare no conflict of interest.

## Abbreviations

The following abbreviations are used in this manuscript:

| | |
|---|---|
| AI | Artificial Intelligence |
| GDPR | General Data Protection Regulation |
| HIPAA | United States' Health Insurance Portability and Accountability Act |
| ECPA | Electronic Communications Privacy Act |
| COPPA | Children's Online Privacy Protection Act |

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
