# Peer review of "Transparency of Artificial Intelligence in Healthcare: Insights from Professionals in Computing and Healthcare Worldwide"

_applsci, doi:10.3390/app122010228_

Round 1
Reviewer 1 Report
The paper discussed what challenges experts and professionals in computing and healthcare identify concerning the transparency of AI in healthcare. The researchers selected five challenges and tried to find the opinions of some experts about them. The idea behind this research is interesting to me however there are some recommendations to improve the paper:
- The researchers should add a section or paragraph to discuss similar research or literature.
- The sample size must be increased to at least 100 participants.
- The researchers did not use any statistical tests to show the results of their study. Examples of such tests are Anova, t-test, regression etc.
- There are some interesting points that the researchers can show such as what is the relation between the experience and the opinion of the participants about specific challenges.
- The discussion section needs some improvement to explain the findings.
- The researchers can move the recommendations to a single section.
Author Response
Paper ID: applsci-1927603
Tittle: Transparency of Artificial Intelligence in Healthcare: Insights From Professionals in Computing and Healthcare Worldwide
Changes based on the reviewers' comments:
First of all, we thank the reviewers for their careful appraisal of our manuscript. Based on their comments, we have reviewed the grammar, spelling, and typos and have addressed each of their points. Our responses to the reviewers’ suggestions and comments are presented below.
Reviewer 1
- State-of-the-Art
Reviewer's comments:
The researchers should add a section or paragraph to discuss similar research or literature.
Response:
We agree with the reviewer and a deep analysis of similar research was added in the following way:
Introduction [Page 01, paragraph 05, line 32]:
Transparency has thus become an urgent and pressing challenge that must be addressed to combat the public's lack of trust in AI [4,32]. This complex and evolving term ``transparency" encompasses in its core issues surrounding the use of AI, such as interpretability, privacy, security, equity, and intellectual property [33].
The scientific community's interest in AI transparency in healthcare is growing at an exponential rate (Fig.1). Aside from recognising the potential and promise that AI has for healthcare, literature reviews in specific study fields are increasingly focusing on outlining barriers that must be surmounted before AI can be reliably deployed and making recommendations to make this happen. Robin [17] argue that communication problems among stakeholders (rationalised in that work in terms of computing and healthcare experts) render the majority of work inaccessible to end users and unlikely to develop further and discuss how a misalignment between open research and the industry's goal in intellectual property protection contributes to the lack of reproducibility and openness. To cope with this situations, they call for engaging clinicians, patients, and relatives in the research process; working towards better representation of global ethnic, socioeconomic, and demographic groups for mitigating equity issues that may end up harming already marginalised and underrepresented groups; and creating and promoting open science initiatives that lead to better reporting, data sharing, and reproducibility, e.g. Human Connectome Project [34]. Mazo et al. [31] provide evidence of the lack of cancer research translation into clinical practise, emphasising that this step remains difficult, in part due to concerns regarding validation and accountability, given that medical mistakes produced by biased or defective AI may threaten patients' lives. On a more optimistic note, the authors note that two projects, the Papanicolau test and Paige Prostate for cervical cancer screening and prostate cancer risk prediction, have recently acquired Food and Drug Administration (FDA) regulatory clearance, a huge step towards real-life deployment. Sajjadian et al. [19] found that, despite the promise of AI in predicting (depression) treatment outcomes, the methodological quality of AI-based investigations is heterogeneous and that those presenting overly optimistic findings are often characterised by inadequate validation. In light of this observation, the authors recommend a more complete assessment of generalisability and replicability, and they advocate using tutorials, consensus criteria, and a checklist to design more effective and deployable AI systems for such purposes.
The common denominator in these and other studies is that there are serious transparency problems that have yet to be addressed, as well as a mismatch between academics' desire to publish research papers showcasing sophisticated AI techniques versus the actual application of such research in healthcare. Stakeholder perspectives on AI trust have also come to the forefront in the literature [28,35,36]. Martinho et al. [28] surveyed 77 medical doctors in The Netherlands, Portugal, and the U.S. about the ethics surrounding Health AI. They found that physicians (i) recognise the need of automating elements of the decision-making process but also state that they must be the ones making the final decision, (ii) acknowledge that, while private tech companies seek profit, it is fundamental to assess AI to the highest standards and hold corporations responsible for any harms caused by their tools, and (iii) highlight the importance of engaging doctors---as end-users---throughout the development process. Esmaeilzadeh [35] collected and analysed responses from 307 individuals in the USA about their perceptions of AI medical devices with clinical decision support characteristics. The authors found that higher concerns, especially regarding performance anxiety and communication barriers, coincide with higher perceived risk and lower benefit, implying that when patients do not comprehend how AI devices work, they distrust them the most. The authors advise carrying out and publishing studies on the benefits and drawbacks of using AI devices such that customers are well aware of them and may have more trust in AI.
Identifying the challenges that professionals in computing and healthcare perceive with AI in healthcare practice might serve as a springboard for additional education, research, and policy development [4,30]. Thus, this position paper seeks to answer: what challenges do experts and professionals in computing and healthcare identify concerning transparency of AI in healthcare? Here we respond to this question based on recent literature discussing this topic and on a global online survey we carried to professionals working in computing and healthcare and potentially within AI. Based on these, we propose a set of recommendations to enhance AI transparency in healthcare.
Bibliography [Page 19]:
- Davenport, T.; Kalakota, R. The potential for artificial intelligence in healthcare.
Future Healthcare Journal 2019, 6, 94–98, https://doi.org/10.7861/futurehosp.6-2-94.
- Borchert, R.; Azevedo, T.; Badhwar, A.; Bernal, J.; Betts, M.; Bruffaerts, R.; Burkhart, M.; Dewachter, I.; Gellersen, H.; Low, A.; et al. Artificial intelligence for diagnosis and prognosis in neuroimaging for dementia; a systematic review. medRxiv 2021. https://doi.org/10.1101/2021.12.12.21267677.
- Sajjadian, M.; Lam, R.W.; Milev, R.; Rotzinger, S.; Frey, B.N.; Soares, C.N.; Parikh, S.V.; Foster, J.A.; Turecki, G.; Müller, D.J.; et al. Machine learning in the prediction of depression treatment outcomes: a systematic review and meta-analysis. Psychological Medicine 2021, p. 1–10. https://doi.org/10.1017/S0033291721003871.
- Martinho, A.; Kroesen, M.; Chorus, C. A healthy debate: Exploring the views of medical doctors on the ethics of artificial intelligence. Artificial Intelligence in Medicine 2021, 121, 102190.
- Manne, R.; Kantheti, S.C. Application of Artificial Intelligence in Healthcare: Chances and Challenges. Current Journal of Applied Science and Technology 2021, 40, 78–89. https://doi.org/10.9734/cjast/2021/v40i631320.
- Mazo, C.; Aura, C.; Rahman, A.; Gallagher, W.M.; Mooney, C. Application of Artificial Intelligence Techniques to Predict Risk of Recurrence of Breast Cancer: A Systematic Review. Journal of Personalized Medicine 2022 12. https://doi.org/10.3390/jpm120
- Heike, F.; Eduard, F.V.; Christoph, L.; Aurelia, T.L. Towards Transparency by Design for Artificial Intelligence. Science and Engineering Ethics 2020, 26, 3333–3361. https://doi.org/10.1007/s11948-020-00276-4.
- Weller, A. Transparency: motivations and challenges. In Explainable AI: Interpreting, Explaining and Visualizing Deep Learning; Springer, 2019; pp. 23–40.
- Van Essen, D.C.; Smith, S.M.; Barch, D.M.; Behrens, T.E.; Yacoub, E.; Ugurbil, K.; Consortium, W.M.H.; et al. The WU-Minn human connectome project: an overview. Neuroimage 2013, 80, 62–79.
- Esmaeilzadeh, P. Use of AI-based tools for healthcare purposes: a survey study from consumers’ perspectives. BMC medical informatics and decision making 2020, 20, 1–19.
- Elemento, O.; Leslie, C.; Lundin, J.; Tourassi, G. Artificial intelligence in cancer research, diagnosis and therapy. Nature Reviews Cancer 2021, 21, 747–752.
- Conclusions
Reviewer's comments:
The sample size must be increased to at least 100 participants.
Response:
We perfectly understand the reviewer's point and are aware of the limited number of respondents in our survey. We made a big effort to increase the number of surveyors without success, and we attribute this situation to the survey fatigue effect experienced during the COVID-19 pandemic. However, this is an invaluable resource considering that we received responses from professionals in healthcare and computing leveraging AI in their routine practice from various countries around the globe.
Due to the need to increase the number of surveyors to make a deeper analysis on those results, we propose this as a future work. All these aspects are reflected in the following way:
Conclusions [Page 15, paragraph 04, line 490]:
Findings from our international survey are somewhat limited by the number of respondents (n=40). We attribute this situation to the survey fatigue effect experienced during the COVID-19 pandemic [46,47]. There are two key aspects to highlight nonetheless. First, we received responses from professionals in healthcare and computing leveraging AI in their routine practice from various countries around the globe. Second, even in this relatively small sample, we were able to note key problems regarding the generalised lack of information available to the general public, of understanding of transparency aspects covered in this work, and of involvement of all stakeholders in the development of AI systems. Future work should consider elaborating on these aspects further, either by means of more respondents or with questionnaires focused on fewer yet deeper items of transparency.
- Results.
Reviewer's comment:
The researchers did not use any statistical tests to show the results of their study. Examples of such tests are Anova, t-test, regression etc.
Response:
We agree with the reviewer that we did not use any statistical test to analyse our results. However, the study was not originally designed for such purposes. We nonetheless took the reviewer’s comment into account and analysed the responses to a subset of the questions stratified by area of expertise and years of experience.
Results [Page 10, paragraph 06, line 290]:
3.7 General perceptions by area of expertise and by years of experience
To further understand whether there were different perceptions depending on the participants' background or experience, we stratified their responses to Q12, Q17, Q18, Q22, Q23, and Q27 by these two factors (Fig. 3a by area and Fig. 3b by experience).
We anticipated that when examining replies by area of expertise, we would see different patterns depending on whether the participants came from a clinical or technological background. However, this was not really the case as both of these groups exhibited similar trends (Fig. 3a): they see favourably privacy and security (Q18, Q22, Q23), have mixed views on interpretability and reporting (Q12 and Q27), and perceive dataset availability and balance between IP rights and transparency unfavourably (Q17 and Q26).
We expected to see experienced professionals with a less favourable opinion of AI in healthcare versus those with less experience due to the relative novelty of AI and its technological, methodological, and ethical requirements and implications for clinical practice. Nonetheless, this does not seem to be the case either (Fig. 3b): participants with more expertise indicated that existing AI solutions met their demands for privacy, security, and patient security (Q18, Q22, and Q23) in general, but those with less years of experience were slightly more doubtful. Furthermore, individuals with fewer years of expertise criticised more AI regarding interpretability, dataset availability, the balance of IP rights and openness, and reporting (Q12, Q17, Q28, and Q27) than the more experienced ones. A relationship between perception and years of experience does not appear to exist in these four elements.
Regardless of expertise and years of experience, there are unmet expectations regarding interpretability, dataset availability, balance between IP rights and transparency, and reporting (Q12, Q17, Q26, and Q27). Worryingly, our findings are consistent with and backed up by ongoing conversations 17,31,35,36,55].
(a) General perception by area of expertise
(b) General perception by years of experience
Figure 3. General perception of the survey respondents stratified by area of expertise and by years of experience.
- Discussion
Reviewer's comment:
There are some interesting points that the researchers can show such as what is the relation between the experience and the opinion of the participants about specific challenges.
Response:
We agree with the reviewer that looking into whether perceptions regarding each of the aspects considered in this work vary with experience is of interest. Accordingly, we modified the results and conclusions accordingly.
Results [Page 10, paragraph 06, line 290]:
3.7 General perceptions by area of expertise and by years of experience
To further understand whether there were different perceptions depending on the participants' background or experience, we stratified their responses to Q12, Q17, Q18, Q22, Q23, and Q27 by these two factors (Fig. 3a by area and Fig. 3b by experience).
We anticipated that when examining replies by area of expertise, we would see different patterns depending on whether the participants came from a clinical or technological background. However, this was not really the case as both of these groups exhibited similar trends (Fig. 3a): they see favourably privacy and security (Q18, Q22, Q23), have mixed views on interpretability and reporting (Q12 and Q27), and perceive dataset availability and balance between IP rights and transparency unfavourably (Q17 and Q26).
We expected to see experienced professionals with a less favourable opinion of AI in healthcare versus those with less experience due to the relative novelty of AI and its technological, methodological, and ethical requirements and implications for clinical practice. Nonetheless, this does not seem to be the case either (Fig. 3b): participants with more expertise indicated that existing AI solutions met their demands for privacy, security, and patient security (Q18, Q22, and Q23) in general, but those with less years of experience were slightly more doubtful. Furthermore, individuals with fewer years of expertise criticised more AI regarding interpretability, dataset availability, the balance of IP rights and openness, and reporting (Q12, Q17, Q28, and Q27) than the more experienced ones. A relationship between perception and years of experience does not appear to exist in these four elements.
Regardless of expertise and years of experience, there are unmet expectations regarding interpretability, dataset availability, balance between IP rights and transparency, and reporting (Q12, Q17, Q26, and Q27). Worryingly, our findings are consistent with and backed up by ongoing conversations 17,31,35,36,55].
(a) General perception by area of expertise
(b) General perception by years of experience
Figure 3. General perception of the survey respondents stratified by area of expertise and by years of experience.
Conclusion [Page 15, paragraph 03, line 479]:
Artificial intelligence has the potential to transform healthcare in the near future. However, in order to be converted into actual practice, these systems must acquire the trust of healthcare professionals and patients, which cannot be done without enhancing their transparency. Understanding the challenges and potential solutions for this purpose is crucial for informing future policies and practices for educating computing and health professionals. Based on current literature and an international survey, we conclude that there is an evident need for: engaging healthcare professionals as these systems are implemented in clinical settings and creating new regulations overseeing the transparent design, validation, deployment, and certification of systems leveraging artificial intelligence. Concerns do not seem particularly linked to the background of the respondents or their years of experience.
Previously, the sentence was [Page 11, paragraph 05, line 403]:
Artificial intelligence has the potential to transform healthcare in the near future. However, in order to be converted into actual practice, these systems must acquire the trust of healthcare professionals and patients, which cannot be done without enhancing their transparency. Understanding the challenges and potential solutions for this purpose is crucial for informing future policies and practices for educating computing and health professionals. Based on current literature and an international survey, we conclude that there is an evident need for: engaging healthcare professionals as these systems are implemented in clinical settings and creating new regulations overseeing the transparent design, validation, deployment, and certification of systems leveraging artificial intelligence.
- Discussion.
Reviewer comment:
The discussion section needs some improvement to explain the findings.
Response:
We agree with the reviewer’s recommendation in terms to improve the findings explanation. We added some graphics and analysis as we detailed for the reviewer’s comments 3 and 4. We decided to keep this new analysis such as part of the Result section.
- Discussion
Reviewer's comments:
The researchers can move the recommendations to a single section.
Response:
We understand the reviewer's proposal to consolidate all recommendations into a single section. Nonetheless, we choose to stick with the present arrangement considering that each of the five aspects explored in this work requires specific recommendations. The work's conclusions provide more broad recommendations.

Reviewer 2 Report
The authors study the application of Artificial Intelligence in health care. They analyze the challenges identified by informatics and health experts and professionals concerning the transparency of AI in health care. They propose five perspectives: interpretability, privacy, security, fairness, and intellectual property. The main problems they focus on include the general lack of information available to the general public, the lack of understanding of the transparency aspects covered in this work, and the lack of participation of all interested parties in developing information systems. AI.
The proposal is very interesting and collects opinions worldwide. Here my comments:
1. The background is poor for an information-gathering job, and many comparative references are missing. Here I suggest some of them:
- Davenport, T., & Kalakota, R. (2019). The potential for artificial intelligence in healthcare. Future healthcare journal, 6(2), 94.
- Del-Valle-Soto, C., Nolazco-Flores, J. A., Puerto-Flores, D., Alberto, J., Velázquez, R., Valdivia, L. J., ... & Visconti, P. (2022). Statistical Study of User Perception of Smart Homes during Vital Signal Monitoring with an Energy-Saving Algorithm. International journal of environmental research and public health, 19(16), 9966.
- Manne, R., & Kantheti, S.C. (2021). Application of artificial intelligence in healthcare: chances and challenges. Current Journal of Applied Science and Technology, 40(6), 78-89.
2. The questionnaire and the sample are interesting, but the authors could better exploit the results obtained—for example, projection graphs or trends towards possible tilts of AI metrics.
3. The classification of the questions is very good, but they could better explain the causes of that classification and why those five angles were chosen. What sources do you have to address this perspective?
4. The conclusions could be broader from the view of applying this classification model for AI.
Author Response
Paper ID: applsci-1927603
Tittle: Transparency of Artificial Intelligence in Healthcare: Insights From Professionals in Computing and Healthcare Worldwide
Changes based on the reviewers' comments:
First of all, we thank the reviewers for their careful appraisal of our manuscript. Based on their comments, we have reviewed the grammar, spelling, and typos and have addressed each of their points. Our responses to the reviewers’ suggestions and comments are presented below.
Reviewer 2
- Bibliography
Reviewer's comments:
The background is poor for an information-gathering job, and many comparative references are missing. Here I suggest some of them:
- Davenport, T., & Kalakota, R. (2019). The potential for artificial intelligence in healthcare. Future healthcare journal, 6(2), 94.
- Del-Valle-Soto, C., Nolazco-Flores, J. A., Puerto-Flores, D., Alberto, J., Velázquez, R., Valdivia, L. J., ... & Visconti, P. (2022). Statistical Study of User Perception of Smart Homes during Vital Signal Monitoring with an Energy-Saving Algorithm. International journal of environmental research and public health, 19(16), 9966.
- Manne, R., & Kantheti, S.C. (2021). Application of artificial intelligence in healthcare: chances and challenges. Current Journal of Applied Science and Technology, 40(6), 78-89.
Response:
We followed the reviewer’s suggestion to increase comparative references. We included the three references recommended by the reviewer and a few extra ones in the latest version of the manuscript. We included the new information in the following way:
We agree with the reviewer, and a deep analysis of similar research was added in the following way:
Introduction [Page 01, paragraph 03, line 23]:
The immense potential of AI in healthcare has also begun to raise serious ethical and legal concerns [4, 25–27]. Healthcare professionals may hesitate, for example, to use the most powerful AI models as these are "black boxes" which decisions and recommendations cannot be fully understood or explained most of the times---not even by computing specialists. Furthermore, establishing the responsibility of AI systems whenever they make mistakes---e.g. due to algorithmic bias---may be difficult, as are the accompanying legal actions [4]. It is difficult for these systems to acquire the trust of healthcare professionals and patients and translate into clinical practice without significant compromise and preparedness of all stakeholders [4,25–31].
Transparency has thus become an urgent and pressing challenge that must be addressed to combat the public's lack of trust in AI [4,32]. This complex and evolving term ``transparency" encompasses in its core issues surrounding the use of AI, such as interpretability, privacy, security, equity, and intellectual property [33].
The scientific community's interest in AI transparency in healthcare is growing at an exponential rate (Fig.1). Aside from recognising the potential and promise that AI has for healthcare, literature reviews in specific study fields are increasingly focusing on outlining barriers that must be surmounted before AI can be reliably deployed and making recommendations to make this happen. Robin [17] argue that communication problems among stakeholders (rationalised in that work in terms of computing and healthcare experts) render the majority of work inaccessible to end users and unlikely to develop further and discuss how a misalignment between open research and the industry's goal in intellectual property protection contributes to the lack of reproducibility and openness. To cope with this situations, they call for engaging clinicians, patients, and relatives in the research process; working towards better representation of global ethnic, socioeconomic, and demographic groups for mitigating equity issues that may end up harming already marginalised and underrepresented groups; and creating and promoting open science initiatives that lead to better reporting, data sharing, and reproducibility, e.g. Human Connectome Project [34]. Mazo et al. [31] provide evidence of the lack of cancer research translation into clinical practise, emphasising that this step remains difficult, in part due to concerns regarding validation and accountability, given that medical mistakes produced by biased or defective AI may threaten patients' lives. On a more optimistic note, the authors note that two projects, the Papanicolau test and Paige Prostate for cervical cancer screening and prostate cancer risk prediction, have recently acquired Food and Drug Administration (FDA) regulatory clearance, a huge step towards real-life deployment. Sajjadian et al. [19] found that, despite the promise of AI in predicting (depression) treatment outcomes, the methodological quality of AI-based investigations is heterogeneous and that those presenting overly optimistic findings are often characterised by inadequate validation. In light of this observation, the authors recommend a more complete assessment of generalisability and replicability, and they advocate using tutorials, consensus criteria, and a checklist to design more effective and deployable AI systems for such purposes.
The common denominator in these and other studies is that there are serious transparency problems that have yet to be addressed, as well as a mismatch between academics' desire to publish research papers showcasing sophisticated AI techniques versus the actual application of such research in healthcare. Stakeholder perspectives on AI trust have also come to the forefront in the literature [28,35,36]. Martinho et al. [28] surveyed 77 medical doctors in The Netherlands, Portugal, and the U.S. about the ethics surrounding Health AI. They found that physicians (i) recognise the need of automating elements of the decision-making process but also state that they must be the ones making the final decision, (ii) acknowledge that, while private tech companies seek profit, it is fundamental to assess AI to the highest standards and hold corporations responsible for any harms caused by their tools, and (iii) highlight the importance of engaging doctors---as end-users---throughout the development process. Esmaeilzadeh [35] collected and analysed responses from 307 individuals in the USA about their perceptions of AI medical devices with clinical decision support characteristics. The authors found that higher concerns, especially regarding performance anxiety and communication barriers, coincide with higher perceived risk and lower benefit, implying that when patients do not comprehend how AI devices work, they distrust them the most. The authors advise carrying out and publishing studies on the benefits and drawbacks of using AI devices such that customers are well aware of them and may have more trust in AI.
Identifying the challenges that professionals in computing and healthcare perceive with AI in healthcare practice might serve as a springboard for additional education, research, and policy development [4,30]. Thus, this position paper seeks to answer: what challenges do experts and professionals in computing and healthcare identify concerning transparency of AI in healthcare? Here we respond to this question based on recent literature discussing this topic and on a global online survey we carried to professionals working in computing and healthcare and potentially within AI. Based on these, we propose a set of recommendations to enhance AI transparency in healthcare.
Bibliography [Page 19]:
- Davenport, T.; Kalakota, R. The potential for artificial intelligence in healthcare.
Future Healthcare Journal 2019, 6, 94–98, https://doi.org/10.7861/futurehosp.6-2-94.
- Borchert, R.; Azevedo, T.; Badhwar, A.; Bernal, J.; Betts, M.; Bruffaerts, R.; Burkhart, M.; Dewachter, I.; Gellersen, H.; Low, A.; et al. Artificial intelligence for diagnosis and prognosis in neuroimaging for dementia; a systematic review. medRxiv 2021. https://doi.org/10.1101/2021.12.12.21267677.
- Sajjadian, M.; Lam, R.W.; Milev, R.; Rotzinger, S.; Frey, B.N.; Soares, C.N.; Parikh, S.V.; Foster, J.A.; Turecki, G.; Müller, D.J.; et al. Machine learning in the prediction of depression treatment outcomes: a systematic review and meta-analysis. Psychological Medicine 2021, p. 1–10. https://doi.org/10.1017/S0033291721003871.
- Martinho, A.; Kroesen, M.; Chorus, C. A healthy debate: Exploring the views of medical doctors on the ethics of artificial intelligence. Artificial Intelligence in Medicine 2021, 121, 102190.
- Manne, R.; Kantheti, S.C. Application of Artificial Intelligence in Healthcare: Chances and Challenges. Current Journal of Applied Science and Technology 2021, 40, 78–89. https://doi.org/10.9734/cjast/2021/v40i631320.
- Mazo, C.; Aura, C.; Rahman, A.; Gallagher, W.M.; Mooney, C. Application of Artificial Intelligence Techniques to Predict Risk of Recurrence of Breast Cancer: A Systematic Review. Journal of Personalized Medicine 2022 12. https://doi.org/10.3390/jpm120
- Heike, F.; Eduard, F.V.; Christoph, L.; Aurelia, T.L. Towards Transparency by Design for Artificial Intelligence. Science and Engineering Ethics 2020, 26, 3333–3361. https://doi.org/10.1007/s11948-020-00276-4.
- Weller, A. Transparency: motivations and challenges. In Explainable AI: Interpreting, Explaining and Visualizing Deep Learning; Springer, 2019; pp. 23–40.
- Van Essen, D.C.; Smith, S.M.; Barch, D.M.; Behrens, T.E.; Yacoub, E.; Ugurbil, K.; Consortium, W.M.H.; et al. The WU-Minn human connectome project: an overview. Neuroimage 2013, 80, 62–79.
- Esmaeilzadeh, P. Use of AI-based tools for healthcare purposes: a survey study from consumers’ perspectives. BMC medical informatics and decision making 2020, 20, 1–19.
- Elemento, O.; Leslie, C.; Lundin, J.; Tourassi, G. Artificial intelligence in cancer research, diagnosis and therapy. Nature Reviews Cancer 2021, 21, 747–752.
Discussion [Page 12, paragraph 01, line 317]:
Because AI-based healthcare systems may have an impact — either positive or negative — on people’s lives, it is fundamental to dispel the myths around these black boxes in order to increase trustworthiness [48,49]. There are some works focus on specific AI techniques in healthcare and interpretability such as common methods, features, training and testing methodologies, metrics, current challenges relating to implementation in clinical practice (these aspects could vary in terms of the project aim, disease, and input/output sources) [51,52]. It is consequently no surprise that interpretability has become a prominent subject in AI for healthcare applications in recent years [32,50–54]. However, it remains an open problem due to current theoretical gaps between model complexity and explainability; insufficient feedback from all stakeholders throughout the development process (e.g. via evaluations of user satisfaction, explanation goodness, acceptance, and trust in AI-based systems); and inadequate traceability indicating how models reach conclusions [32,55]. According to our international survey, a fundamental issue that needs to be addressed is the unequal understanding among professionals: most healthcare professionals do not understand how AI systems operate nor whether these adhere to medical guidelines. However, participants also manifested the level of granularity is essential. While a certain degree of clarity of the whole process — equations, biography, medical guideline, and algorithms — seems reasonable, providing excessive details may be confusing and of no practical value. We propose the following recommendations for better interpretability:
Previously, the sentence was [Page 06, paragraph 08, line 243]:
Because AI-based healthcare systems may have an impact — either positive or negative — on people’s lives, it is fundamental to dispel the myths around these black boxes in order to increase trustworthiness [48,49]. It is consequently no surprise that interpretability has become a prominent subject in AI for healthcare applications in recent years [32,50–54]. However, it remains an open problem due to current theoretical gaps between model complexity and explainability; insufficient feedback from all stakeholders throughout the development process (e.g. via evaluations of user satisfaction, explanation goodness, acceptance and trust in AI-based systems); and inadequate traceability indicating how models reach conclusions [32,55]. According to our international survey, a fundamental issue that needs to be addressed is the unequal understanding among professionals: most healthcare professionals do not understand how AI systems operate nor whether these adhere to medical guidelines. However, participants also manifested the level of granularity is essential. While a certain degree of clarity of the whole process — equations, biography, medical guideline, and algorithms — seems reasonable, providing excessive details may be confusing and of no practical value. We propose the following recommendations for better interpretability:
- Results
Reviewer's comment:
The questionnaire and the sample are interesting, but the authors could better exploit the results obtained—for example, projection graphs or trends towards possible tilts of AI metrics.
Response:
We agree with the reviewer. We took the reviewer’s comment into account and analysed the responses to a subset of the questions stratified by area of expertise and years of experience.
Results [Page 10, paragraph 06, line 290]:
3.7 General perceptions by area of expertise and by years of experience
To further understand whether there were different perceptions depending on the participants' background or experience, we stratified their responses to Q12, Q17, Q18, Q22, Q23, and Q27 by these two factors (Fig. 3a by area and Fig. 3b by experience).
We anticipated that when examining replies by area of expertise, we would see different patterns depending on whether the participants came from a clinical or technological background. However, this was not really the case as both of these groups exhibited similar trends (Fig. 3a): they see favourably privacy and security (Q18, Q22, Q23), have mixed views on interpretability and reporting (Q12 and Q27), and perceive dataset availability and balance between IP rights and transparency unfavourably (Q17 and Q26).
We expected to see experienced professionals with a less favourable opinion of AI in healthcare versus those with less experience due to the relative novelty of AI and its technological, methodological, and ethical requirements and implications for clinical practice. Nonetheless, this does not seem to be the case either (Fig. 3b): participants with more expertise indicated that existing AI solutions met their demands for privacy, security, and patient security (Q18, Q22, and Q23) in general, but those with less years of experience were slightly more doubtful. Furthermore, individuals with fewer years of expertise criticised more AI regarding interpretability, dataset availability, the balance of IP rights and openness, and reporting (Q12, Q17, Q28, and Q27) than the more experienced ones. A relationship between perception and years of experience does not appear to exist in these four elements.
Regardless of expertise and years of experience, there are unmet expectations regarding interpretability, dataset availability, balance between IP rights and transparency, and reporting (Q12, Q17, Q26, and Q27). Worryingly, our findings are consistent with and backed up by ongoing conversations 17,31,35,36,55].
(a) General perception by area of expertise
(b) General perception by years of experience
Figure 3. General perception of the survey respondents stratified by area of expertise and by years of experience.
- Introduction
Reviewer's comments:
The classification of the questions is very good, but they could better explain the causes of that classification and why those five angles were chosen. What sources do you have to address this perspective?
Response:
Transparency is a complex construct that evades simple definitions, there is not a single correct way to define it. We followed the guidelines used by Heike, F. and Weller, A., which include, but are not restricted to, explainability, interpretability, openness, accessibility, and visibility.
We summarize and set the main topics into five groups --- interpretability, privacy, security, equity, and intellectual property --- which were used in our paper. We considered a way to articulate the topics in understandable terms, which can often be framed as answering questions of “what”, “how”, or “why”. This is clarify in the following way:
Introduction [Page 01, paragraph 05, line 32]:
Transparency has thus become an urgent and pressing challenge that must be addressed to combat the public's lack of trust in AI [4,32]. This complex and evolving term ``transparency" encompasses in its core issues surrounding the use of AI, such as interpretability, privacy, security, equity, and intellectual property [3]. In this work, we understand transparency from these five viewpoints.
Previously, the sentence was [Page 01, paragraph 05, line 29]:
The notion of ``transparency" has appeared as an alternative to combat the lack of trust in AI [28]. In its core, transparency is a broad term encompassing issues surrounding the use of AI, such as interpretability, privacy, security, equity, and intellectual property [29]. Nonetheless, this is a complex, evolving, and ambiguous term that requires maturation [29]. Identifying the challenges that professionals in computing and healthcare perceive with AI in healthcare practice might serve as a springboard for additional education, research, and policy development.
- Conclusion
Reviewer's comments:
The conclusions could be broader from the view of applying this classification model for AI.
Response:
We are unsure of what the reviewer meant by “applying this classification model for AI”. In this work, we do not construct a “classification model” nor intent to “apply it” as we seek to identify challenges that professionals in computing and healthcare perceive based on recent literature as well as a global online survey that we carried out to professionals working in these two fields.

Round 2
Reviewer 1 Report
All the comments have been addressed
Reviewer 2 Report
Thanks to the authors for the resolved comments. The manuscript would be ready for publication.